# Design and Control of Soft-Rigid Grippers for Food Handling

Valerio Bo[1,2], Leonardo Franco[1], Enrico Turco[1,2], Maria Pozzi[1,2], Monica Malvezzi[1],
Domenico Prattichizzo[1,2] and Gionata Salvietti[1,2]

*Abstract*— **Food handling is a challenging task for robotic grippers, as it requires to manipulate highly deformable and fragile items, that can be easily damaged. Moreover, ingredients for the preparation of the different dishes are usually stored in small containers that are often not easily accessible. This abstract introduces design and control techniques to design soft-rigid grippers. Specifically, we propose an innovative soft-rigid, tendon-driven gripper: the Double-Scoop Gripper (DSG). Its two-fingered design exploits a specialized structure to cope with constrained spaces. The DSG can delicately grasp objects of various shapes by employing two scoop-shaped fingertips that can form a single plate when fingers are flexed. DSG capabilities are verified with experiments conducted using real food ingredients within a pick-and-place setup to evaluate both the grasping and the releasing capability of the gripper. Obtained results are promising and suggest that this approach could be particularly advantageous in the context of automated food serving.**

## I. INTRODUCTION

Food manufacturing has witnessed significant advancements in automation and robotics, revolutionizing food handling processes. In this context, industrial food service robots play a pivotal role in enhancing efficiency, conserving space, and elevating cleanliness and safety standards [1].

Over the past few decades, robotics found widespread usage in tasks like dispensing ingredients, executing precise cuts, packaging or casing food items, as well as skillfully picking and placing products into containers while also facilitating the sorting process [2]. Food handling is an important task also in assistive applications in which robot manipulators can be used to help people in the kitchen or feed and serve patients with upper limb impairments [3]. When dealing with pick-and-place tasks, robotics must face two main challenges [4]. To begin with, the robot end-effector must delicately handle the fragile nature of food items susceptible to damage. Secondly, the robot capability to adjust to environmental changes is crucial, ideally without necessitating a reconfiguration of the entire robotic setup.

In this abstract, we propose a collection of methods for controlling and designing soft-rigid grippers, demonstrating their usefulness in a real world scenario. Leveraging our knowledge of environmental constraint exploitation (ECE)

We gratefully acknowledge the support of the European Union by the HORIZON project "HARIA - Human-Robot Sensorimotor Augmentation" (GA No. 101070292) and by the Next Generation EU project ECS00000017 "Ecosistema dell'Innovazione" Tuscany Health Ecosystem (THE, PNRR: Spoke 9 - Robotics and Automation for Health).

[1]Universita degli Studi di Siena, Dip. di Ingegneria dell'Informazione e Scienze Matematiche, Siena, Italy. {name.surname}@unisi.it

[2]Istituto Italiano di Tecnologia, Genoa, Italy {name.surname}@iit.it

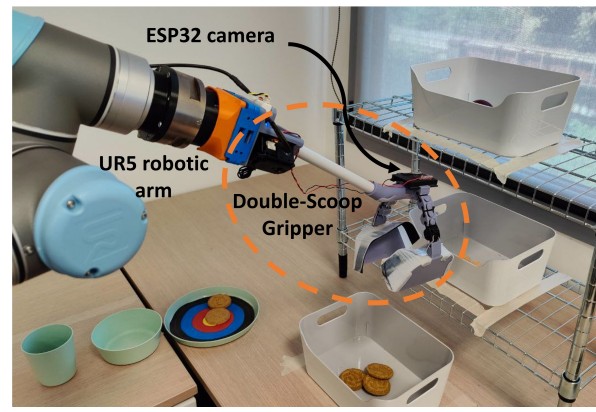

Fig. 1: Experimental setup for food handling. Pick area: 3 containers at different heights (table, within and above the shelves). Place area: 3 containers of varying sizes (plate, bowl, glass). The proposed Double-Scoop Gripper, embedding an on-board camera, is indicated with an orange circle.

[5] and using prior experience in both control and design methods in this area [6]–[8], we developed a novel gripper, the Double-Scoop Gripper (DSG), specifically designed for pick-and-place scenarios (Fig. 1). Leveraging the presence of environmental constraints in food tray assembly scenarios, we positioned at the tips of these fingers a flat, rigid component able to slide below the food items with additional flexible borders that create a scoop. This unique configuration offers a threefold advantage: firstly, it allows the gripper to cage objects rather than applying conventional pinching; secondly, it improves the releasing of the food items with respect to other soft grippers; lastly, it enables the gripper to grasp the food exploiting the constraints present in the environment. The gripper palm is attached to the robot arm end-effector through a rigid support element to cope with narrow environments.

The proposed gripper was tested evaluating the grasping and releasing success rate in a scenario with real ingredients in multiple containers and a tray, considering single and multiple objects pick-and-place operations.

## II. METHODOLOGY

### A. Design Techniques for Scoops

To develop soft-rigid grippers, we propose design methods incorporating data-driven techniques by analyzing precedent grasping experiences. Thus, the design becomes intelligent, embodying information of precedent grasp planning. Doing

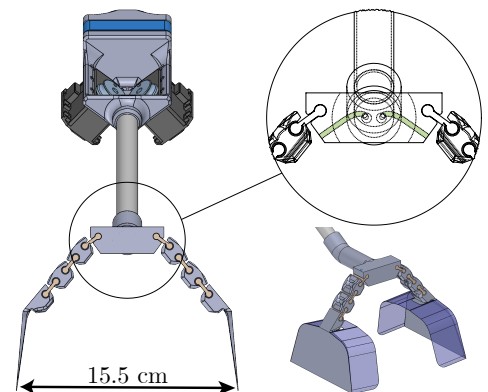

Fig. 2: CAD design of the Double-Scoop Gripper.

so can ease the grasp of planning complexity because part of it has already been considered during the design. First, we developed a data-driven framework to merge the optimal design and planning of the scoops [6]. This is done by evaluating the optimal positioning and size of the scoop. In this analysis, we considered the presence of the environment and consequently planned grasps, orienting the scoop parallel to it, implicitly making the grasps generation for the execution more straightforward. Then, we decided to develop a framework that guarantees their mechanical stability while reducing the amount of material to be printed [8]. We evaluated the interaction between the scoop and different objects through simulation, and then we translated these force signals into instructions for Topology Optimization.

Thus, we developed a novel soft-rigid gripper, the Double-Scoop Gripper, integrating elements from the proposed methods. The core components of the gripper are the two scoops placed at the fingertips. They are designed to slide under food items, avoiding damage. The two scoops have a surface area of $60 \, cm^2$ each. We also added soft protection walls to the three sides of the scoop to force the food to stay on the gripper base. Another essential feature of our design is the use of tendon-driven actuation through a rigid plastic pipe characterized by a length of $20 \, cm$. To actuate the fingers and redirect the tendons' path, we design a unique structure, highlighted in Fig. 2, shaped to reduce the friction on the tendon, avoiding the addition of pulleys. These design features allowed us to significantly reduce the gripper bulkiness, making it possible to grasp objects in narrow spaces.

All the rigid parts are made with Elegoo ABS-like resin except for the tube, which is a commercial PVC tube. All the flexible parts are made of 85A shore TPU. We used two Dynamixel MX-28AT motors controlled by an ESP32 microcontroller connected to a custom transistor-transistor logic adapter board.

### B. Control Methods for Environmental Constraints Exploitation

To deal with environmental constraints while using scoops, we developed optimization and machine learning techniques. First, we explored executing grasps with the scoop, considering the gripper features and the environment around the

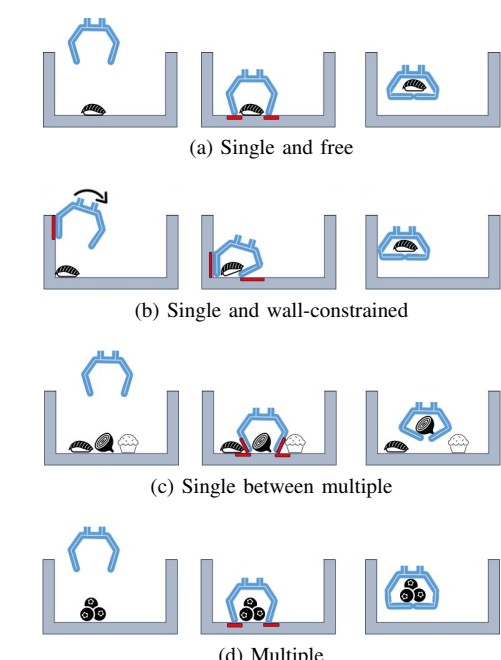

(a) Single and free

(b) Single and wall-constrained

(c) Single between multiple

(d) Multiple

Fig. 3: Picking strategies depending on the placement and amount of food items. Grey: container walls, blue: gripper, red: areas where the gripper exploits the environment.

manipulandum by solving an optimization problem [7]. We developed a grasping strategy called scoop grasp, where the scoop slides onto surfaces in contact with the target object. We devised this strategy, including cases in which objects are constrained from two sides. Then, we proposed a data-driven methodology to observe and replicate how humans would approach the objects with the scoop. We developed a novel Learning from Demonstrations (LfD) method that is faster in computing feasible grasps and needs a little training data [9]. We also considered different representations of the objects when extracting the grasping primitives to generalize the method.

Thus, we developed different grasping strategies based on these methods depending on where the food is placed inside a container. The approaches have been determined considering the part of the container exploited by the scoops. We identified 4 strategies: *i*) unconstrained single quantity of food; *ii*) single food item close to the walls of the container; *iii*) single target food among multiple food items; *iv*) multiple target food items close to each other. Fig. 3 shows a sketch of the strategies.

These strategies demonstrate the effectiveness of the gripper in exploiting the environment around single and multiple items, also considering a target food among numerous. In all the strategies, the opening of the scoops is adjusted according to the size of the food item. Besides, when exploiting the walls, the gripper is tilted to place the scoop closer to the lateral surface to adapt to it.

Following similar reasonings for the placing, we implemented three placing strategies, considering containers of different sizes (Fig. 4). The identified strategies highlight the

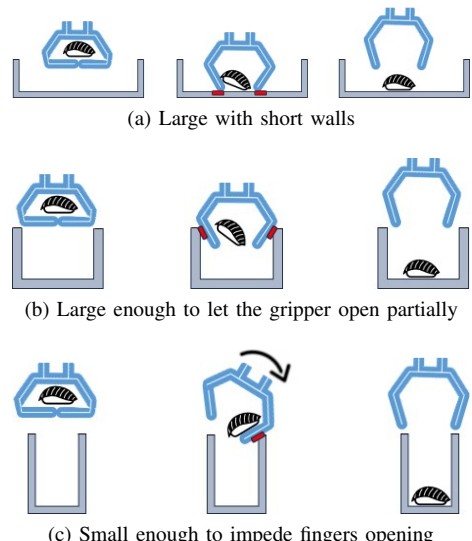

(a) Large with short walls

(b) Large enough to let the gripper open partially

(c) Small enough to impede fingers opening

Fig. 4: Placing strategies depending on the size of the target container. Grey: container walls, blue: gripper, red: areas where the gripper exploits the environment.

ability of the gripper to adapt to the containers by exploiting their different characteristics. If the release occurs in a large container, we exploit the environment on the bottom surface, while the scoops are laid on the container lateral walls if the container walls are tight.

## III. RESULTS AND DISCUSSION

### A. Experimental setup

During the experimental phase, the Double-Scoop Gripper was mounted on a UR5 robotic arm. To test the performance of the DSG, we conducted a pick-and-place procedure, evaluating separately the results of the two tasks. The aim was to demonstrate that the proposed gripper can grasp and release single and multiple objects by applying the proposed strategies. The picking scenario shown in Fig. 1 consists of grabbing food from 3 containers placed at different heights. In the placing scenario, instead, the task was to release the food in containers of different sizes, such as a plate, a bowl, and a glass. For both tasks, we employed 5 real food items of different sizes, shapes, and softness (see Fig. 5). We tested each scenario with all the objects 5 times each, collecting 100 and 150 trials for pick and place, respectively.

The results in terms of success rates for the conducted experiments are shown in Table I, where the denominators of the success rates are evaluated by multiplying the number of trials (25 for each case) by the number of tested food items ($n = 1$ or $n = 3$).

Regarding the picking experiments, we evaluated the four picking conditions for each object. A grasp was considered successful if the target food was held inside the gripper until it arrived above the releasing container. Table I reports the results of the picking phase (from the second to the sixth column). Overall, the DSG successfully grasped 133 out of 150 objects, obtaining a success rate of about $88.6\%$. In

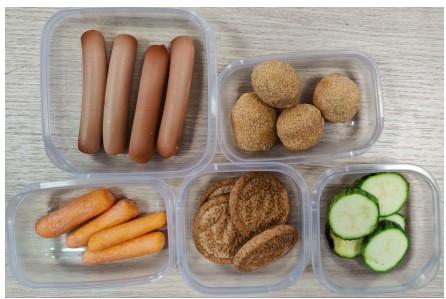

Fig. 5: Food items adopted in the experimental trials: sausages, meatballs, carrots, cookies, and zucchini.

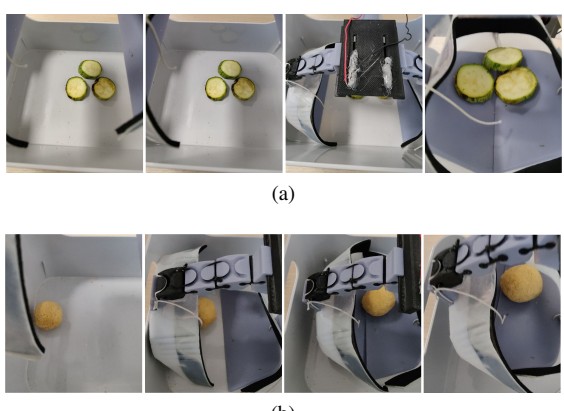

(a)

(b)

Fig. 6: Sequences of picks. a) Picking of multiple zuchini; b) successful wall-pick of a meatball.

particular, a single quantity of food placed away from the container walls was picked-up in $96\%$ of the trials, whereas, in constrained cases, it was caught in $68\%$. Instead, when the target food was placed among others, $88\%$ of grasps were successful. Lastly, multiple quantity of the same object ($n = 3$) placed away from the walls were grasped in the $94.6\%$ of cases. Almost all objects were easily grasped when placed away from the constraints of the container (see Fig. 6a). The only exception is cookies, which were grabbed $80\%$ and $86.6\%$ in the single and multiple cases, respectively. On the other hand, grasping becomes quite challenging when the container wall constrains an object. Meatballs are the only items the DSG was able to grab in all the trials (see Fig. 6b), proving the efficacy of the developed strategy and the ability of the gripper to exploit the constraint, coping with narrow environments.

Regarding the placing experiments, $97\%$ of objects were successfully placed, as shown in the last column of Table I. In more detail, the DSG released all the food items in the plate with a success rate of $100\%$ in both single and multiple cases. The same result was obtained when the bowl was the release container. On the other hand, releasing foods into the glass was the most challenging. The gripper successfully placed $92\%$ and $90.6\%$ of food items into the glass, in single and multiple quantities of objects, respectively. Most of the releases were successful for all the objects (see Fig. 7). The most challenging food item to release was the zucchini,

TABLE I: Pick-and-place results. Columns 2-6 contain the results from the picking strategies, while columns 7-13 show the results from the placing. $n$ indicates the number of target food items. The overall results of the two experimental phases are represented in bold.

| Food items | Pick | | | | | Place | | | | | | |
|---|---|---|---|---|---|---|---|---|---|---|---|---|
| | **Free** $n=1$ | **Constrained** $n=1$ | **Target** $n=1$ | **Free** $n=3$ | **All** (item) | **Plate** $n=1$ | $n=3$ | **Bowl** $n=1$ | $n=3$ | **Glass** $n=1$ | $n=3$ | **All** (item) |
| Meatballs | 5/5 | 5/5 | 5/5 | 15/15 | 30/30 | 5/5 | 15/15 | 5/5 | 15/15 | 5/5 | 13/15 | 58/60 |
| Cookies | 4/5 | 3/5 | 3/5 | 13/15 | 23/30 | 5/5 | 15/15 | 5/5 | 15/15 | 5/5 | 13/15 | 58/60 |
| Carrots | 5/5 | 4/5 | 5/5 | 15/15 | 28/30 | 5/5 | 15/15 | 5/5 | 15/15 | 5/5 | 15/15 | 60/60 |
| Sausages | 5/5 | 2/5 | 5/5 | 14/15 | 26/30 | 5/5 | 15/15 | 5/5 | 15/15 | 5/5 | 15/15 | 60/60 |
| Zucchinis | 5/5 | 3/5 | 4/5 | 14/15 | 26/30 | 5/5 | 15/15 | 5/5 | 15/15 | 3/5 | 12/15 | 55/60 |
| All (strategy) | 24/25 | 17/25 | 22/25 | 71/75 | **133/150** | 25/25 | 75/75 | 25/25 | 75/75 | 23/25 | 68/75 | **291/300** |

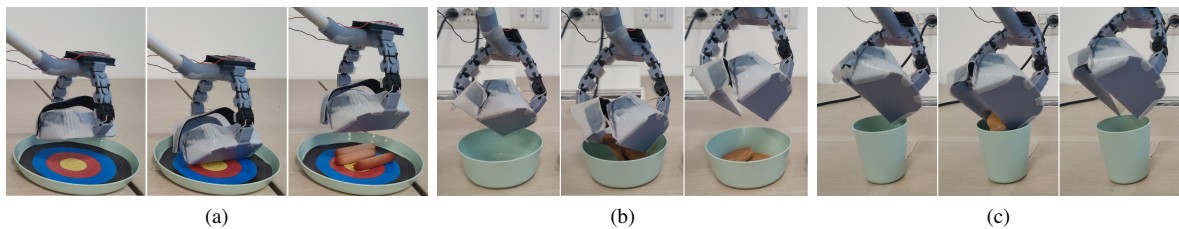

(a)         (b)         (c)

Fig. 7: Sequence of multiple placement. a) Sausages in the plate; b) cookies in the bowl; c) meatballs in the glass.

which presented $91.6\%$ of successful placements. Moreover, the zucchini was the only item that showed a failure in the single placement. The glass was the only container where multiple objects failed. The issue was caused by the tightness of the container itself, which hindered the placement of more than two items when their size was notable.

## IV. CONCLUSION

In this abstract, we proposed design and control techniques to develop a novel soft-rigid gripper, the Double-Scoop Gripper. This gripper is specifically tailored to tackle food tray assembly scenarios, where food is placed in restricted spaces. The gripper structure comprises two opposing tendon-driven soft-rigid structures with a rigid component at their tips. To deal with narrow environments, the gripper is situated at the end of a pipe and is equipped with an RGB camera.

Four picking and three placing strategies were developed to fully exploit the gripper characteristics. The DSG was tested in pick-and-place trials to validate its design. The experiments showed that the DSG is able to comply with the food fragility without damages, and to reach containers in narrow environments. The gripper achieved a high success rate in both grasp and release tasks, exploiting the environment in different conditions.

In some of the tests, we noticed possible limitations of the proposed design, but these can be overcome by modifying the fingers structure to obtain a better adaptation of the scoops to the environment. Future research will also focus on implementing a prismatic joint to substitute the actual fixed pipe to increase the range of motion of the robotic arm and achieve configurations that previously were unfeasible.

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
