# OpenReview forum: "Design and Control of Soft-Rigid Grippers for Food Handling"
_IEEE.org/2024/ICRA/Workshop/CookingRobot — CookingRobot2024 Poster_

### Official Review · Reviewer_rRY7 · 2024-04-11
**The review of "Design and Control of Soft-Rigid Grippers for Food Handling"**

**Rating:** 7
**Confidence:** 4

**Review:**

This paper proposes  soft-rigid, tendon-driven gripper: the Double-Scoop Gripper (DSG) for food handling.
The mechanism enables the gripper to grasp the food exploiting the constraints present in the environment.

Major Comments
* The paper is well organized and the gripper mechanism is novel. Also the picking strategies depending on the placement and amount of food items, and the placing strategies depending on the size of the target container are very nice.
* Although there are no comparative experiments, the performance is being evaluated in various situations.

Video
* It is very helpful to understand the motion of the gripper.

---

### Official Review · Reviewer_kV4f · 2024-04-16
**The review for "Design and Control of Soft-Rigid Grippers for Food Handling"**

**Rating:** 7
**Confidence:** 3

**Review:**

This paper proposes the Double-Scoop Gripper (DSG), a soft-rigid tendon-driven gripper that pick-and-place food. Food handling is a very important issue and this innovative hand proposal will enhance the discussion at the workshop.

Major comment:
* It would be nice to have a discussion of the advantages and disadvantages of the DSG compared to other food handling hands, if not an actual comparison experiment.

Video:
* The video is very good, showing the 4 picking strategies and the 3 placing strategies, and how the hand is able to pick while exploiting the constraints present in the environment.